# A Two-Stage Auction Mechanism for 3PL Supplier Selection under Risk Aversion

**Fuqiang Lu [1], Hualing Bi [1,\*], Yanli Hu [2], Wenjing Feng [2], Suxin Wang [2] and Xu Zhang [1]**

[1] School of Economics and Management, Yanshan University, Qinhuangdao 066004, China; fuqiang_lu@126.com (F.L.); 15232366482@163.com (X.Z.)

[2] Faculty of Information Science and Engineering, Northeastern University, Shenyang 810819, China; huyanli2020@163.com (Y.H.); fwj15530620815@163.com (W.F.); wsx96@126.com (S.W.)

\* Correspondence: bihualing081@126.com

**Abstract:** The third party logistics (3PL) suppliers selection is a key issue in sustainable operation of fourth party logistics (4PL). A two-stage auction mechanism is designed for the selection of 3PL suppliers. Different from previous studies, the paper considers risk preference of 4PL integrators during the auction and uses the prospect theory to establish the auction scoring function of 4PL integrators. First, a first score sealed auction (FSSA) mechanism is used to solve the selection problem. However, the results show that FSSA is not an ideal method. Hence, the English auction (EA) mechanism is combined with the FSSA mechanism to form a two-stage auction. The FSSA is taken as the first stage auction, and the EA is taken as the second stage auction, and the two-stage auction mechanism is constructed. The two-stage auction can improve the utility of the 4PL integrator and the auction efficiency. In addition, for the degree of disclosure of attribute weights in the scoring function, two states, complete information and incomplete information is designed. In case analysis, the validity of the designed two-stage auction mechanism is verified. The 4PL integrator can obtain higher utility under the risk-neutral auction than the risk-averse auction. The complete information auction does not make the 4PL integrator obtain higher utility than the incomplete information auction.

**Keywords:** sustainable operation; supplier selection; two-stage auction; risk aversion; information disclosure

## 1. Introduction

The fourth party logistics is a new type of logistics operation mode developed on the basis of the development of third-party logistics practice in order to achieve higher efficiency, more professional, and more integrated and sustainable logistics services. The fourth party logistics effectively integrates social logistics resources and creates value by influencing the entire supply chain. It not only controls and manages specific logistics services, but also proposes a planning solution for the entire logistics process to meet the diverse needs of customers and sustainable goal of supply chain management [1]. In actual operation, the fourth party logistics is carried out by third party logistics companies to carry out specific logistics activities. Therefore, choosing a 3PL supplier is a key issue in 4PL management and very important to maintaining sustainable operations of logistics businesses and supply chains. A large number of commercial practices have proved that when there are multiple suppliers with limited competition opportunities, the purchaser can use the reverse auction mechanism to select suppliers, which can improve the procurement efficiency and reduce the procurement cost while ensuring the suppliers' fair competition.

In the past, research on logistics service supply chain is common for all participants to use their own maximum benefit, i.e., risk neutrality, as the criterion for decision-making. However, in practice, the behaviors of decision-makers are often influenced by psychological preferences. Decision-makers generally have risk preference attitudes. Therefore, considering the behavioral characteristics of members in the supply chain, they can fully reflect the supply chain management behavior in reality. Risk preference is generally expressed as risk aversion at the collective and individual levels [2]. Therefore, this paper considers the risk aversion preference psychology in the 4PL integrator's auction decision. The prospective theory [3,4] can well describe the behavioral characteristics of risk aversive people. The theory holds that individuals will measure gains and losses before making decisions. Individuals are more sensitive to loss than income.

By considering both the risk attributes and the attributes under a commercial criterion, this paper designs a novel two-stage auction mechanism for supplier selection based on multi-attribute auction and sustainable supply chain management. In the first stage, a multi-auction mechanism is established to determine the shortlist among all qualified suppliers based on two attributes (price and delivery time) under a commercial criterion. In the second stage, risk attributes against the shortlisted suppliers are further considered, and a new ranking method based on grey correlation degree of mixed sequence is proposed to rank the finalists and to select the final winners.

The main contributions of this paper are as follows: (1) The previous works assume that both buyers and sellers are risk neutral, this paper considers the risk aversion of 4PL integrator and introduces the prospect theory into the auction scoring function; (2) designing a two-stage auction combining the first scoring seal auction with the English Auction (EA), for the promotion of 4PL integrator auction utility; (3) for the degree of disclosure of attribute weights in the scoring function, designs, and compares the two auction methods in aspect of complete information auction and incomplete information auction.

The remainder of the paper is structured in three sections. Section 2 presents a literature review; FSSA mechanism and a case analysis are given in Section 3; Section 4 presents a two-stage auction mechanism and some analysis on it; in Section 5, the conclusions, defect, and outlook are provided.

## 2. Literature Review

### 2.1. Auction Mechanism and Information Disclosure

Auction is a market mechanism that participates in bidding according to rules to achieve resource allocation and price [5]. In recent years, the research on auction bidding is as follows. Mithas [6] studied the effect of bidding competition, information asymmetry, reserve price, bid decrement, auction duration, and bidder type on buyer surplus. He collected field data on more than 700 online procurement auctions conducted by a leading auctioneer and involving procurement items worth millions of dollars. He found that bid decrement and auction duration have no effect on B2B procurement auctions. The results suggest that the use of the rank-bidding format increases buyer surplus when incumbent suppliers participate in the auction. Zeng [7,8] studied the bidding strategies of bidders in multi-attribute reverse auctions and the expected returns of both parties. Ding [9] designed an auction mechanism based on buyer preference disclosure for the private information of buyers and sellers in the auction, and analyzed how the seller used the evaluation model to bid. Pan [10] combined multi-attribute utility theory with probability theory to construct a multi-attribute procurement model, which proved the optimal bidding strategy of suppliers. Yu [11] proposes a negotiation protocol special for multi-product supplier selection problem. The negotiation protocol is a hybrid multi-agent protocol of combinatorial procurement auction protocol and multi-bilateral bargaining protocol. The negotiation protocol is able to support the purchasing company and suppliers negotiate on the concrete commitments of multiple products simultaneously, and select suppliers

for multiple products. Simulation is conducted to demonstrate the effectiveness and efficiency of the negotiation protocol. Rao [12] investigates the problem of supplier selection under multi-source procurement for a type of divisible goods (such as coal, oil, and natural gas). Alaei [13] proposed a combinatorial reverse auction mechanism to select suppliers for the required items of a company. As a contribution, it is assumed that the task of supplying each required item is indivisible to multiple suppliers, or the company prefers to select only one supplier for supplying each required item. So, the winner determination process is done in such a way that supplying each tendered item is assigned to only one potential supplier.

In real auctions, 4PL integrator need to hide some of the information in order to protect their own interests, so that they do not disclose all the information during the auction. In previous studies, the types of information disclosed in the auction and the degree of disclosure of the information will have different effects on the results, which reflects the complexity and necessity of information disclosure in auction research. In the auction, 4PL integrator have a certain weight bias for each attribute. This is the preference information of 4PL integrator. This paper takes into account whether the degree of disclosure will affect the auction results. Wilson [14] began to study the impact of information in auctions. Liu [15] proposed that the disclosure of information during the auction process will affect the bidder's bargaining behavior and thus affect the bidding results. In response to how to effectively disclose information, Zhu [16] designed an auction mechanism to encourage bidders to disclose more auction information to achieve optimal auction. Zhang, Gong [17–19] et al. explored effective strategies for information disclosure in multi-attribute reverse auctions. Rothkopf [20] and others proposed that the relevant research on information disclosure in bidding will become an important research content.

### 2.2. Risk Aversion in Supplier Selection

In recent years, risk aversion has been considered in the study of supplier selection problem [21,22], and many kinds of methods have been applied. Maskin and Rile [23] studied the buyer's optimal auction problem under risk aversion and the seller's risk neutrality. Qian [24] proposed a winner decision model based on cumulative prospect theory for the buyer's risk aversion behavior in reverse auction. Huang [25] proposed the PT-BOCR model to solve the problem of risk aversive buyers choosing suppliers in the auction. Xu [26] proposed risk-averse optimal bidding strategy based on VaR for demand-side resources under uncertain conditions. Li [27] considers that the choice of risk response schemes for complex equipment research and manufacturing is a consensus issue of group negotiation. The paper exploits group decision-making and utility theory to establish a risk disposal scheme selection model for complex equipment development based on group negotiation consensus, and then a case verifies the validity and rationality of the proposed model. The results show that the consensus scheme selection problem proposed in the paper effectively combines the preference value and utility, considers the supplier's risk preference behavior, and achieves the multisubject consensus scheme. Chen [28] considers supplier selection from the perspective of risk aversion. They proposed generalized intuitionistic fuzzy soft set (GIFSS) combined with extending gray relational analysis (GRA) method to select an appropriate supplier from the perspective of risk aversion in group decision-making environment. Finally, a numerical example for supplier selection is given to illustrate application of the method, and the comparisons with other methods are also made. Dupont [29] studies the problem of supplier selection and order allocation in a retail supply chain under disruption risk. The risk sensitivity of the decision-maker is considered and a mixed integer linear programming approach to provide decision-making support that shows a supply manager the "elasticity of (expected) losses versus (expected) profits". Alikhani [30] considers factors like sustainability and suppliers' risk factors into the supplier selection problem, simultaneously, and proposes a multi-method approach based on quantitative empirical investigations, and analytical modeling. The model is developed for both risk-neutral and risk-averse decision-makers. The efficiency

and applicability of the proposed framework is demonstrated through a real case. Results show that considering sustainability criteria or risk factors separately results in inappropriate decisions. Yu [31] develops a novel integrated supplier selection approach incorporating the decision-maker's risk attitude using the ANN, AHP, and TOPSIS methods. The decision-maker's risk attitude toward procurement transaction is originally considered in the supplier selection process. In the proposed approach, the ANN model is used to classify the decision-maker's risk attitude. Elham [32] proposes a mixed-integer non-linear programming (MINLP) model for the integrated supplier selection and order allocation in a centralized supply chain considering the disruption risks and a risk-averse decision-maker. In the latter case, they apply two types of risk assessment tools introduced in the finance literature to analyze the decision-maker's behavior: value-at-risk (VaR) and conditional value-at-risk (CVaR).

## 3. FSSA Mechanism

### 3.1. Auction Mechanism Design

The FSSA is a multi-attribute auction form that expands the traditional first-price sealed auction. The traditional first-price sealed auction only uses the highest price as the criterion for judging the auction winner, the first scoring seal auction is based on the highest auction scores that combine multiple attributes as the criteria for judging the winner of the auction [33]. The FSSA has the advantage of effectively countering the collusion behavior of the suppliers and facilitating the participation of new bidders.

4PL integrator undertakes customer's logistics project orders, provides services to customers from the perspective of customers, and 3PL suppliers complete the implementation of logistics. There are multiple 3PL suppliers involved in the competition, and the 4PL integrator chooses one to complete the customer's logistics project order. Since delivery date delays and cost overruns are common in logistics projects, 3PL suppliers are considered for transaction price and delivery date. In addition, 4PL integrators and 3PL suppliers are two different interests. 4PL integrators expect short delivery date and lower prices, while 3PL suppliers expect to have as much time as possible and can pay higher prices. In order to balance the interests of both, designing an auction mechanism can protect the interests of 4PL integrator and the interests of 3PL suppliers. In addition, considering subjective psychological factors, in general, decision-makers cannot be absolute risk neutral, and generally tend to risk aversion, so 4PL integrators have risk aversion and risk neutrality to make decisions and compare.

#### 3.1.1. Scoring Function

In the auction, the preference between 4PL integrator and 3PL suppliers is reflected by the scoring function. The 4PL integrator evaluates the quality of 3PL suppliers by the scoring function.

In the process of human decision-making, the decision-making body is considered to be completely rational, which is an idealized hypothesis that does not conform to the actual situation. The decision-making subject is emotionally influenced by the emotional decision-making process, which will make the final decision-making not the most effective rational decision, foreground theory. PT [3,4] is the theory of individual decision-making behavior under uncertainty conditions, describing the irrational behavior in behavioral decision-making. In this paper, the PT [23,24] is introduced to construct the scoring function.

(1) Scoring function of auction under risk aversion

In this paper, two attributes, price and delivery date are considered. It is assumed that the two attributes are independent of each other, that is, there is no correlation between the attributes. Therefore, the scoring function of the 4PL integrator is the sum of the utility of the two attributes in the PT, as shown in Formula (1). Variables are defined in table 1, which will be used in the following formulas.

**Table 1.** Variable description.

| Variable | Meaning |
| --- | --- |
| $d$ | Bidding delivery date of 3PL supplier |
| $q$ | Bidding price of 3PL supplier |
| $d_{max}$ | The maximum delivery date, expressed as the delivery date reservation value $r_d$ in an auction |
| $q_{max}$ | The highest price, expressed as the price reservation value $r_q$ in an auction |
| $d_{min}$ | The minimum delivery date of 3PL supplier |
| $q_{min}$ | The lowest price of 3PL supplier |
| $p_d$ | Weight of delivery date given by 4PL integrator |
| $p_q$ | The price weight given by the 4PL integrator |
| $q_0$ | Expected price of 4PL integrator |
| $d_0$ | Expected delivery date of 4PL integrator |
| $p_d^1$ | Weight of delivery date given by 3PL supplier |
| $p_q^1$ | Weight of price given by 3PL supplier |
| $q_0^1$ | Expected price of 3PL supplier |
| $d_0^1$ | Expected delivery date of 3PL supplier |

Price and delivery date are cost type attributes for 4PL integrator. The smaller the price and delivery date of the transaction, the bigger the utility, so the computation of profit and loss value is $\Delta\pi=\pi_0-\pi$. Formula (1) is the scoring function of 4PL integrator under risk aversion.

$$U_1 = w_q v(\Delta\pi_q) + w_d v(\Delta\pi_d) \tag{1}$$

where,
$$v(\Delta\pi_q)=\begin{cases} \Delta\pi_q^{\alpha} & \Delta\pi_q \geq 0 \\ -\lambda(-\Delta\pi_q)^{\beta} & \Delta\pi_q < 0, \quad \lambda \geq 1 \end{cases},$$

$$v(\Delta\pi_d)=\begin{cases} \Delta\pi_d^{\alpha} & \Delta\pi_d \geq 0 \\ -\lambda(-\Delta\pi_d)^{\beta} & \Delta\pi_d < 0, \quad \lambda \geq 1 \end{cases}, \quad \Delta\pi_q = q_0 - q \quad , \quad \Delta\pi_d = d_0 - d \quad ,$$

$$w_q^+ = \frac{p_q^{\gamma}}{(p_q^{\gamma}+(1-p_q^{\gamma}))^{1/\gamma}} \quad , \quad w_q^- = \frac{p_q^{\delta}}{(p_q^{\delta}+(1-p_q^{\delta}))^{1/\delta}} \quad , \quad w_d^+ = \frac{p_d^{\gamma}}{(p_d^{\gamma}+(1-p_d^{\gamma}))^{1/\gamma}} \quad ,$$

$$w_d^- = \frac{p_d^{\delta}}{(p_d^{\delta}+(1-p_d^{\delta}))^{1/\delta}} .$$

(2) Scoring function of auction under risk neutral

The risk neutral behavior of 4PL integrator in auction decision-making is described by changing the parameter $\alpha=\beta=\gamma=\delta=\lambda=1$ [34] in PT, that is, the scoring function of auction under risk neutral is Formula (1), where, $\Delta\pi_q = q_0 - q$ , $\Delta\pi_d = d_0 - d$ ,

$v(\Delta\pi_q)=\Delta\pi_q$ , $v(\Delta\pi_d)=\Delta\pi_d$ , $w_q = p_q$ , $w_d = p_d$ .

(3) Point Estimation of Interval Probability

Interval probability theory (IPT) is an important theoretical method for studying uncertainty decision-making and risk type decision-making. In classical probability theory,

point-value forms are used to describe the probability of occurrence of events, but in IPT, probabilities are expressed in interval form, and classical probability is a special form of interval probability. The interval probability should satisfy two conditions. If the interval probability is $p_j = (p_j^-, p_j^+)$, it should satisfy the rationality, that is, the existence of the interval probability $p_j$ should be satisfied $0 \le p_j^- \le p_j \le p_j^+ \le 1$ $\sum_{j=1}^{n} p_j = 1$. In addition, it should satisfy the feasibility of satisfying both regularity and non-negativity $\sum_{j=1}^{n} p_j^- \le 1$ $\sum_{j=1}^{n} p_j^+ \ge 1$.

The nonlinear transformation method, i.e., the maximum entropy method, is an effective and important interval probability point estimation method. It was proposed by Jaynes [35] in 1957. The maximum entropy method requires the estimation of the natural state probability distribution under the current information-only condition. The main idea is to select the distribution with the largest entropy as the estimation of the state space distribution among all the distributions that meet the known conditions, which is widely used in decision analysis [36,37]. The entropy maximization criterion solving problem can generally be described as finding the largest entropy distribution on the closure of a given distribution set. When it is a convex set, its solution exists and is unique [38], it actually solves the following optimal planning problem: $\max H(x) = -\int_{x \in \Theta} f(x) \ln f(x) dx$, the constraint is $\int_{x \in \Theta} f(x) dx = 1$, $\int_{x \in \Theta} f(x) g_i(x) dx = E_i, i = 1, 2, \cdots, n$. In the formula, $\Theta$ is the domain of $x$, $f(x)$ is an unknown probability density function, $g_i(x)$ represents known function information.

In order to obtain a point estimate of the interval probability, an optimization model can be established. Let the interval probability $p = [p_j^-, p_j^+]$ be obtained by calculating the following optimization model Formulas (2)–(4), and get its point estimate $\hat{p}_j$.

$$\max H(p) = -\sum_{j=1}^{n} \hat{p}_j \ln \hat{p}_j \tag{2}$$

s.t.

$$\sum_{j=1}^{n} \hat{p}_j = 1 \tag{3}$$

$$0 \le p_j^- \le \hat{p}_j \le p_j^+ \le 1 \tag{4}$$

### 3.1.2. Scoring Function of 3PL Supplier

3PL suppliers use their scoring function to judge the utility of their bid information. In order to reasonably maximize their own interests. The scoring function of 3PL suppliers is private. This paper only considers the case that 3PL suppliers are risk neutral. The scoring function of 3PL suppliers is the same as the risk neutral 4PL integrator's scoring function. However, on the contrary to 4PL integrator, price and delivery date are beneficial attributes for 3PL suppliers. If the final price and delivery datea deal are bigger then, its utility is bigger. Therefore, the profit and loss value is calculated as $\Delta\pi = \pi - \pi_0$. The scoring function of 3PL supplier is shown in Formula (5).

$$U_2 = w_q^1 v(\Delta\pi_q) + w_d^1 v(\Delta\pi_d) \tag{5}$$

where, $\Delta\pi_q = q - q_0^1$, $\Delta\pi_d = d - d_0^1$. $v(\Delta\pi_q) = \Delta\pi_q$, $v(\Delta\pi_d) = \Delta\pi_d$, $w_q^1 = p_q^1$, $w_d^1 = p_d^1$.

### 3.1.3. Utility Functions of 4PL Integrator

4PL integrator have different scoring functions under different risk attitudes. How can 4PL integrator objectively evaluate 3PL suppliers without considering risk attitude? This requires a utility function that is not affected by risk attitude. It is private information for 4PL integrator and is not available to 3PL suppliers.

Multi-attribute utility theory (MAUT) is adopted to establish the utility function of 4PL integrator [39]. Its measurement does not take into account the risk attitude and auction methods of 4PL integrator. It is shown in Formula (6).

$$U_4 = p_q \frac{q_{\max} - q}{q_{\max} - q_{\min}} + p_d \frac{d_{\max} - d}{d_{\max} - d_{\min}} \tag{6}$$

The evaluation function of each attribute is given by using the normalization method for different attributes, and then the value of the evaluation function is added up by simple weighting method to obtain the utility function of the 4PL integrator.

### 3.1.4. Bidding Model of 3PL Suppliers

Each 3PL supplier performs a one-time bidding based on the auction model. The 3PL bidding model enables 3PL suppliers to bid for 4PL integrator auction requirements and 3PL suppliers cost requirements, and maximize their own interests. The auction model of the 3PL supplier is shown in Equations (7) and (8).

$$(p_i, d_i) = \arg\max U_2^i \prod_{i \neq j}^{n-1} prob(U_1^i > U_1^j) = \arg\max U_2^i F^{n-1}(U_1^i) \tag{7}$$

s.t.

$$U_1 \sim U(a,b) \tag{8}$$

$$d_{\min} \leq d \leq d_{\max} \tag{9}$$

$$q_{\min} \leq q \leq q_{\max} \tag{10}$$

Equation (7) represents the price and delivery date for the 3PL supplier bid to maximize its expected utility. That is, while the 4PL integrator scores the highest for the 3PL supplier (the 3PL supplier becomes the auction winner), the 3PL supplier's own score is maximized (making the 3PL supplier as high utility as possible). Equation (8) represents a uniform distribution in which the score $U_1$ of the 4PL integrator to the 3PL supplier satisfies the interval ($a,b$). For the lowest score $a$, each 3PL supplier bid has the lowest score for the reservation value. For the highest score $b$, each 3PL supplier bids at its own cost score for the highest score of the auction. Equation (9) represents that the delivery date should meet the constraints of its own delivery date cost and auction delivery date reservation value. Equation (10) represents that the price should meet the constraints of its own price cost and auction price reservation value. That is, the cost is used as the lower limit of the bid, and the auction threshold is used as the higher limit of the bid.

### 3.1.5. Auction Process

The specific implementation process of the auction is designed as follows.

Step 1: The 4PL integrator first publishes the auction information, including the expected value of the price, the reserved value of the price, the weight of the price, expected value of delivery date, the reserved value of the delivery date, the weight of the delivery date, scoring function, the number of participants in the auction of 3PL suppliers.

Step 2: All 3PL suppliers submit their own bid information in one time.

Step 3: The 4PL integrator selects the 3PL supplier with the highest score as the auction winner, and the winner provides the item or service with the submitted bid information, and the auction ends.

*3.2. Cases*

This section verifies the effectiveness of the designed auction model through a case and analyzes the influence of risk attitude on auction results. Suppose that a 4PL integrator with risk attitude selects 3PL suppliers by auction approach for logistics project cooperation, and attributes like price and delivery date are considered in the auction.

3.2.1. Initial Auction Information

(1) Initial auction information of 4PL integrator

4PL integrator releases the auction information, expected price of 4PL integrator is $0.35 m, $q_0 = 350,000$, conservative price is $0.39 m $r_q = 390,000$, weight of price $p_q = 0.45$, expected delivery date $d_0 = 200$ day, conservative delivery date $r_d = 230$ day, weight of delivery date $p_d = 0.55$, the number of 3PL suppliers is 10.

(2) Initial auction information of 3PL suppliers

It assumes that the scoring functions of 3PL suppliers are private information, the weight of price and delivery date for each 3PL suppliers are $p_q^1 = 0.5, p_d^1 = 0.5$, respectively. The expected price and delivery date of 3PL suppliers are the mean of their cost value and reservation value respectively. The reservation values are $r_q = 390$, $r_d = 230$.

The minimum delivery date and lowest price of each 3PL supplier are shown in Table 2. As long as the transaction delivery date and price are no less than their minimum delivery date and lowest price respectively, 3PL suppliers will continue to participate in the auction.

**Table 2.** Cost information of 3PL suppliers.

| 3PL Suppliers | 1 | 2 | 3 | 4 | 5 | 6 | 7 | 8 | 9 | 10 |
|---|---|---|---|---|---|---|---|---|---|---|
| Minimum delivery date | 202 | 198 | 202 | 196 | 198 | 195 | 199 | 193 | 188 | 185 |
| The lowest price | 340 | 350 | 335 | 348 | 349 | 352 | 341 | 353 | 358 | 362 |

The principle on design of cost information for 3PL suppliers, is that if one attribute is higher, another attribute will be lower, and the difference of the values under the same attribute is not very big. The purpose is to limit the advantages of 3PL supplier, and so as to not cover up the influence of 4PL integrator on auction results under different risk attitudes and auction methods.

The highest score of each 3PL supplier can be calculated based on the information in Table 2, which is unknown to the 4PL integrator and other 3PL suppliers, but only for the purpose of analysis in this paper. According to the scoring function (1) in Section 3.1.1, calculate the highest score of 3PL supplier under risk aversion. The results are shown in Table 3.

**Table 3.** The highest score of 3PL supplier under risk aversion.

| 3PL Suppliers | 1 | 2 | 3 | 4 | 5 | 6 | 7 | 8 | 9 | 10 |
|---|---|---|---|---|---|---|---|---|---|---|
| highest score | 0.016 | 0.041 | 0.066 | 0.112 | 0.061 | 0.005 | 0.129 | 0.027 | −0.031 | 0.022 |

By comparing the data in Table 3, it is concluded that 3PL supplier 7 has the highest score and 3PL supplier 9 has the lowest one under risk aversion.

According to the scoring function (2) in Section 3.1.1, the 3PL supplier's highest score is compared under risk neutral of 4PL integrator, and the results are shown in Table 4.

**Table 4.** The highest score of 3PL supplier under risk neutral.

| 3PL Suppliers | 1 | 2 | 3 | 4 | 5 | 6 | 7 | 8 | 9 | 10 |
|---|---|---|---|---|---|---|---|---|---|---|
| cost value | 0.076 | 0.037 | 0.132 | 0.096 | 0.048 | 0.069 | 0.12 | 0.095 | 0.13 | 0.14 |

By comparing the data in Table 4, it is concluded that 3PL supplier 10 has the highest score and 3PL supplier 2 has the lowest one under risk neutral.

3.2.2. Analysis of Experiments Results

(1) Analysis of auctions results under risk aversion

Each 3PL supplier bids according to the bidding model in Section 3.1.4, and the bidding results are shown in Table 5. The bids for all 3PL suppliers in Table 5 are all equal to the expected value of the 4PL integrator.

**Table 5.** Bidding results of 3PL suppliers under risk aversion.

| 3PL Suppliers | 1 | 2 | 3 | 4 | 5 | 6 | 7 | 8 | 9 | 10 |
|---|---|---|---|---|---|---|---|---|---|---|
| delivery date | 200 | 200 | 200 | 200 | 200 | 200 | 200 | 200 | 200 | 200 |
| price | 350 | 350 | 350 | 350 | 350 | 350 | 350 | 350 | 350 | 350 |

According to the scoring function (1) in Section 3.1.1, calculate the bid score of each 3PL supplier, which is shown in Table 6. In this case, the 4PL integrator is risk aversion, the bid scores of all 3PL suppliers are 0, which means that, it is unable to determine an auction winner.

**Table 6.** Score for 3PL supplier bidding.

| 3PL Suppliers | 1 | 2 | 3 | 4 | 5 | 6 | 7 | 8 | 9 | 10 |
|---|---|---|---|---|---|---|---|---|---|---|
| Highest score | 0 | 0 | 0 | 0 | 0 | 0 | 0 | 0 | 0 | 0 |

(2) Analysis of auctions results under risk neutral

Each 3PL supplier bids according to the bidding model in Section 3.1.4, and the bidding results are shown in Table 7. According to the scoring function (5), the bid scores for each 3PL supplier are calculated and shown in Table 8.

**Table 7.** Bidding of 3PL suppliers under 4PL integrator with risk neutral.

| 3PL Suppliers | 1 | 2 | 3 | 4 | 5 | 6 | 7 | 8 | 9 | 10 |
|---|---|---|---|---|---|---|---|---|---|---|
| delivery date | 206.1 | 198 | 209.2 | 196 | 198 | 195 | 199 | 193 | 188 | 185 |
| price | 340 | 353.3 | 335 | 356.5 | 353.3 | 358.1 | 351.6 | 361.4 | 369.6 | 374.4 |

**Table 8.** Bid scores for 3PL supplier.

| 3PL Suppliers | 1 | 2 | 3 | 4 | 5 | 6 | 7 | 8 | 9 | 10 |
|---|---|---|---|---|---|---|---|---|---|---|
| Highest score | $-6.7 \times 10^{-5}$ | $-8.3 \times 10^{-6}$ | $8.3 \times 10^{-5}$ | $-1.7 \times 10^{-5}$ | $-8.3 \times 10^{-6}$ | $3.17 \times 10^{-4}$ | $-4.17 \times 10^{-6}$ | $8.3 \times 10^{-5}$ | $-5 \times 10^{-5}$ | $5 \times 10^{-4}$ |

The 4PL integrator is risk neutral. In Table 8, 3PL supplier 10 has the highest cost value among the 3PL suppliers, $5 \times 10^{-4}$, and becomes the auction winner. The auction mechanism helps the 4PL integrator to select the correct 3PL supplier, which indicates the effectiveness of the designed auction mechanism. But, for the winner, the auction score $5 \times 10^{-4}$ is much smaller than the 3PL supplier's cost value 0.14 (see Table 4). Therefore, the 3PL suppliers has great space to continue the bid, and for 4PL integrator, there is chance to improve in his utility.

## 4. Two-Stage Auction Mechanism

### 4.1. Auction Mechanism Design

The results of the first score sealed auction (FSSA) is not good enough to solve the 3PL supplier selection problem. Therefore, we would like to combine FSSA with the English auction (EA) to form a two-stage auction mechanism, which can make the 4PL integrator's auction score higher. Perform the second stage of the EA after the first stage of FSSA, and the starting score of the EA is the highest score of the FSSA. The two-stage auction mechanism is shown in Figure 1.

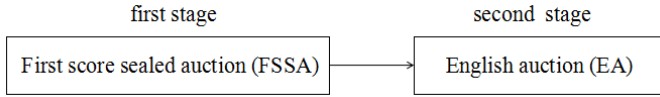

**Figure 1.** Two-stage auction mechanism.

Most of the two-stage auctions are based on negotiation and auction [40]. In some works of two-stage auctions, first select the suppliers who meet the requirements by negotiation, and then obtain the final winner by auction. In other works, the winner is determined by auction first, and then negotiate the final results [41]. Studies on two-stage auctions have also achieved some development [40–43], different auction forms are used in the two stages. Moreover, the two-stage auction has been successfully applied to NTL Broadband Cable Co.'s acquisition of Cablelink Limited, Ireland's largest cable provider, the sale of the Hong Kong government and Los Angeles land, and the Fujian Property Rights Exchange's stake in the Sedrin Beer [43].

### 4.1.1. Process of EA

The specific implementation process of the EA is designed as follows.

Step 1: The 4PL integrator publishes the auction information, including the expected value of the price, the reserved value of the price, the weight of the price, expected value of delivery date, the reserved value of the delivery date, the weight of the delivery date, scoring function, the number of 3PL suppliers participation in the auction.

Step 2: 4PL integrator randomly assigns bidding serial number to 3PL supplier before bidding. Here, it assumes that the number of the first 3PL supplier is 1, the number of the second 3PL supplier is 2, and so on. 3PL suppliers bid in sequence by the serial number.

Step 3: 4PL integrator determines the minimum bid score increment $D$, that is, the bid score of the later 3PL supplier should be at least $D$ higher than that of the previous 3PL supplier. Then gradually reduce the value of $D$ depending on the specific situation.

Step 4: Each 3PL supplier will bid according to the serial number, until no 3PL supplier can bid. The current 3PL supplier with the highest score will be the final winner. The price and delivery date obtained from the winner are the final transaction price and delivery date of the auction.

### 4.1.2. EA Bidding Model for 3PL Supplier

The 3PL supplier will bid according to the bidding model, which is shown in Formulas (11) and (12).

$$MaxU_2 \tag{11}$$

s.t.

$$U_1 \geq U_1^{former} + D \tag{12}$$

The objective function (11) is to maximize the score of the 3PL suppliers. Formula (12) indicates that the biding score of 4PL integrator should be higher than the bidding

score from the last 3PL supplier $U_1^{former}$. $D$ is the minimum bid score increment determined by 4PL integrator [44,45], where $U_1$ is determined by risk attitude and auction mechanism, and constraints (9) and (10) are also needed to be met.

*4.2. Cases*

4.2.1. Initial Auction Information

The initial auction information for 4PL integrator and 3PL suppliers is the same as in Section 3.2.1.

4.2.2. Analysis of Experiments Results

(1) Analysis of the auctions results under risk aversion

Under risk aversion of 4PL integrator, in first-stage, FSSA has the same score for each 3PL supplier, and in the second stage, EA starts with the starting score 0.1. The second-stage auction process is shown in Table 9.

**Table 9.** The second-stage auction process under risk aversion.

| Auction Number | 3PL Supplier | Bidding Information (Price, Delivery Date) | Score | Continue Bidding? |
|---|---|---|---|---|
| 1 | 1 | -- | 0.1 | No |
| 2 | 2 | -- | 0.1 | No |
| 3 | 3 | -- | 0.1 | No |
| 4 | 4 | (348.3196) | 0.1 | Yes |
| 5 | 5 | -- | 0.11 | No |
| 6 | 6 | -- | 0.11 | No |
| 7 | 7 | (342.8199) | 0.11 | Yes |
| 8 | ... | -- | 0.12 | No |

(2) Analysis of the auctions results under risk neutral

Under risk neutrality of 4PL integrator, in first stage, 3PL supplier 10 has a maximum score 5e−4, and in the second stage the EA starts with 0.1. The second-stage auction process is shown in Table 10. Summary of the auction results under different risk attitude is shown in Table 11.

**Table 10.** The second-stage auction process under risk neutrality.

| Auction Number | 3PL Supplier | Bidding Information (Price, Delivery Date) | Score | Continue Bidding? |
|---|---|---|---|---|
| 1 | 1 | -- | 0.1 | No |
| 2 | 2 | -- | 0.1 | No |
| 3 | 3 | (335,203.8) | 0.1 | Yes |
| 4 | 4 | -- | 0.11 | No |
| 5 | 5 | -- | 0.11 | No |
| 6 | 6 | -- | 0.11 | No |
| 7 | 7 | (341.9199) | 0.11 | Yes |
| 8 | 8 | -- | 0.12 | No |
| 9 | 9 | (358.9188) | 0.12 | Yes |
| 10 | 10 | (362.9185) | 0.13 | Yes |
| 11 | ... | -- | 0.14 | No |

**Table 11.** Auction results under different risk attitude.

| 4PL Integrator Attitude | 3PL Supplier Winner | Transaction Results | 4PL Integrator Utility |
|---|---|---|---|
| Risk aversion | 7 | (342.8199) | 0.7651 |
| Risk neutral | 10 | (362.9185) | 0.7717 |

In Table 11, the 3PL supplier with the highest score under the current risk attitude is selected by the two-stage auction. The cost score of 3PL supplier 7 is the highest when the 4PL integrator is risk averse, this result is same as in Table 3. The cost score of 3PL supplier 10 is the highest when the 4PL integrator is risk neutral, this result is same as in Table 4. So, the effectiveness of the two-stage auction is verified.

When the 4PL integrator is risk averse, the highest score of 3PL supplier under FSSA is 0 (see Table 6), and the highest score of 3PL supplier under two-stage auction is 0.7651. When the 4PL integrator is risk neutral, the highest score of the 3PL supplier under FSSA is $5 \times 10^{-4}$ (see Table 8), and the highest score of the 3PL supplier under the two-stage auction is 0.7717. Hence, the two-stage auction obtained a higher score than the simple FSSA, that is, the 4PL integrator gained higher utility under two-stage auction.

Comparing the two-stage auction with EA, the starting score in the EA is the score corresponding to the reserved point published by the 4PL integrator. When the delivery date reservation value is 230 and the price reservation value is 390, the initial score under risk aversion can be obtained according to Equation (1), which is −2.045. According to Equation (2), the initial score under risk neutral is −1. The two-stage auction starts at 0.1, the 4PL integrator can design the starting score of the second stage EA more accurately because of the FSSA in the first stage, so the auction round is reduced, and the 4PL integrator obtains higher auction efficiency. The two-stage auction combines the advantages of the two auction methods, 4PL integrator can choose the 3PL supplier by first performing the FSSA and then the EA.

Table 11 also shows that the utility of 4PL integrator under risk neutral is 0.7717, which is higher than the utility of 4PL integrator under risk averse, which is 0.7651. Because when 4PL integrator auction is under risk aversion, for the same profit and loss value, 4PL integrator will psychologically magnify the negative utility of losses, thus concealing the positive utility brought by benefits. 4PL integrator are not psychologically allowed to have expected losses, that is, requirements $d \leq 200, q \leq 350$. Although 4PL integrator did not feel the loss, but in fact, they ignored part of the benefits, 4PL integrator shows irrational behavior. Therefore, 4PL integrator maintains a risk neutral rational attitude in auction, which can bring higher utility to itself.

4.2.3. Analysis on Incomplete Attribute Weight Information

This section considers a situation of incomplete information in the auction process, specifically, the attribute weights in the 4PL integrator's scoring function are considered. The attribute weight truly reflects the preference of the 4PL integrator. If the information is not fully disclosed, the core information of the customer can be effectively protected from the leak, effectively preventing the false suppliers who are not satisfied with the business from exploring the business information in the name of auction cooperation. Moreover, the degree of disclosure of attribute weight information directly affects the bids of 3PL suppliers and thus affects their score [40]. Therefore, this paper studies the incomplete disclosure of attribute weight information. Different from the complete information auction, that the attribute weight information is fully announced. Under the incomplete information situation, the attribute weight of the 4PL integrator is reserved for the 3PL supplier, which is released in the form of interval. At the same time, the risk aversion of the 4PL integrator is still considered.

Based on surveys of enterprises and literatures [40], the initial data about the delivery date weight and price weight are designed. Usually, the weights are value of points under

the situation of complete information. However, the auction is under the situation of incomplete information, the values of the weights are not determined values or are uncertain values. Therefore, this section applies a form of interval to describe this kind of uncertainty, and then applies Formulas (2)–(4) to calculate the values of the delivery date weight and price weight, respectively. According to the published attribute weight range of 4PL integrator from small to large, three specific cases are designed to test the influence of the incomplete information on the auction results. These specific cases are called Case I, II, III, respectively.

(1) Case I

Assume that the 4PL integrator publishes the delivery date weight and price weight in the range $w_d = (0.5, 0.6)$, $w_q = (0.4, 0.5)$. Then the 4PL integrator announces a large delivery date weight and price weight $w_d = (0.54, 0.6)$, $w_q = (0.44, 0.5)$, then the 4PL integrator announces a small delivery date weight and price weight $w_d = (0.5, 0.56)$, $w_q = (0.4, 0.46)$, as shown in Figure 2.

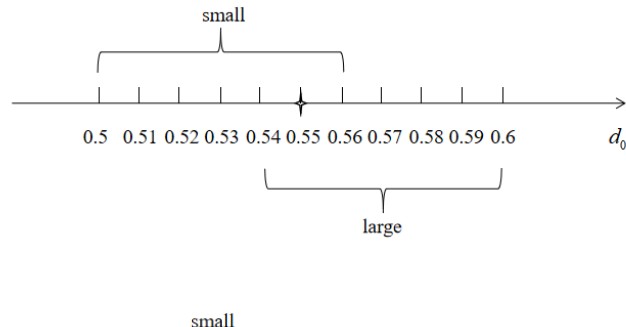

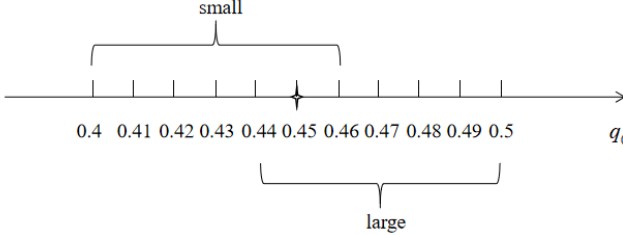

**Figure 2.** Interval of attribute weight information (case I).

Then according to Formulas (2)–(4), the combination of these four intervals can generate the point weight as $w_d = 0.53, 0.55, 0.57$, $w_q = 0.43, 0.45, 0.47$.

(2) Case II

Assume that the 4PL Integrator publishes the delivery date weight and price weight in the range $w_d = (0.4, 0.7)$ $w_q = (0.3, 0.6)$. Then the 4PL integrator announces a large delivery date weight and price weight $w_d = (0.5, 0.7)$ $w_q = (0.4, 0.6)$, then the 4PL integrator announces a small delivery date weight and price weight $w_d = (0.4, 0.6)$ $w_q = (0.3, 0.5)$, which is shown in Figure 3.

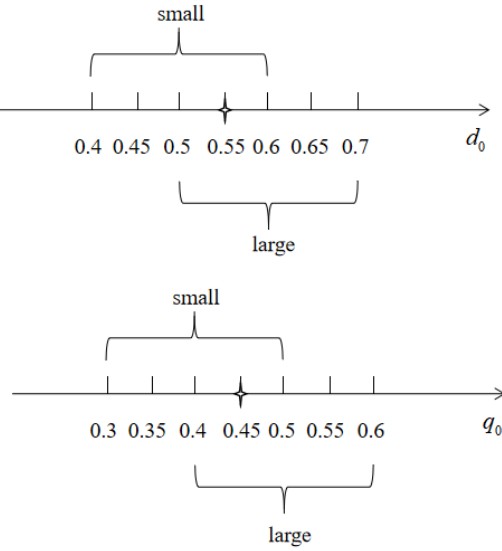

**Figure 3.** Interval of attribute weight information (case II).

Then according to Formulas (2)–(4), the combination of these four intervals can generate the point weight $w_d = 0.5, 0.55, 0.6$, $w_q = 0.4, 0.45, 0.5$.

(3) Case III

Assume that the 4PL integrator publishes the delivery date weight and price weight in the range $w_d = (0.3, 0.8)$, $w_q = (0.2, 0.7)$. Then the 4PL integrator announces a large delivery date weight and price weight $w_d = (0.5, 0.8)$, $w_q = (0.4, 0.7)$, then the 4PL integrator announces a small delivery date weight and price weight $w_d = (0.3, 0.6)$ $w_q = (0.2, 0.5)$, which is shown in Figure 4.

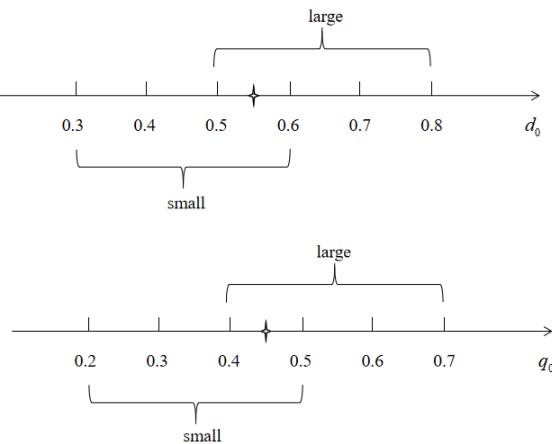

**Figure 4.** Interval of attribute weight information (case III).

Then according to Formulas (2)–(4), the combination of these four intervals will estimate the point weight $w_d = 0.45, 0.55, 0.6, 0.65$, $w_q = 0.35, 0.4, 0.45, 0.55$.

For the above estimated point weights, the two-stage auction is implemented. The results of calculation are divided into two parts: (1) The auction results under risk aversion; (2) the auction results under risk neutrality.

### 4.2.4. Comparative Analysis

The experimental results for the three cases are compared and summarized in Table 12. It can be seen from Table 12 that the auction winner is the 3PL supplier with the highest cost score under the current 4PL integrator's risk attitude. The 3PL supplier 7 is the winner under risk aversion of 4PL integrator, which also can be found in Table 3. The 3PL supplier 10 is the winner under risk neutral of 4PL integrator, which also can be seen in Table 4.

**Table 12.** Comparison of auction winners under different risk attitude.

| $(w_d, w_q)$ | (0.65,0.35) | (0.6,0.4) | (0.57,0.43) | (0.55,0.45) | (0.53,0.47) | (0.5,0.5) | (0.45,0.55) |
|---|---|---|---|---|---|---|---|
| Risk aversion | 7 | 7 | 7 | 7 | 7 | 7 | 7 |
| Risk neutral | 10 | 10 | 10 | 10 | 3 | 3 | 3 |

The transaction price and the delivery date of the 3PL supplier winner are summarized in Table 13.

**Table 13.** Transaction results of 3PL supplier winners under different risk attitude.

| $(w_d, w_q)$ | (0.65,0.35) | (0.6,0.4) | (0.57,0.43) | (0.55,0.45) | (0.53,0.47) | (0.5,0.5) | (0.45,0.55) |
|---|---|---|---|---|---|---|---|
| Risk aversion | (341.9199) | (342.4199) | (342.6199) | (342.8199) | (342.9199) | (343.2199) | (342.7199.5) |
| Risk neutral | (365.4185) | (363,185) | (362.6185) | (362.9185) | (335,202.6) | (335,204.7) | (335,205.1) |

Formula (6) is used to calculate the utility of the 4PL integrator under risk aversion, and Table 14 is obtained. The results in Table 14 are also shown in Figure 5.

**Table 14.** Utility comparison of 4PL integrator under risk aversion.

| $(w_d, w_q)$ | (0.65,0.35) | (0.6,0.4) | (0.57,0.43) | (0.55,0.45) | (0.53,0.47) | (0.5,0.5) | (0.45,0.55) |
|---|---|---|---|---|---|---|---|
| Risk aversion | 0.7724 | 0.7683 | 0.7667 | 0.7651 | 0.7643 | 0.7618 | 0.7598 |

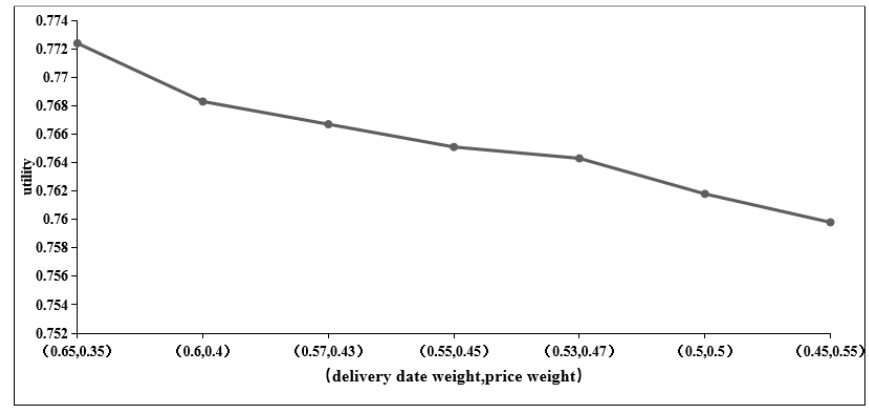

**Figure 5.** The utility of 4PL integrator under risk aversion.

The highest utility of 4PL integrator for risk neutral auctions is obtained at $w_d = 0.65$, $w_q = 0.35$.

Formula (6) is used to calculate the utility of the transaction result in the risk-neutral 4PL integrator, and Table 15 is obtained. The results in Table 15 are shown in Figure 6.

**Table 15.** Utility comparison of 4PL integrator under risk neutral.

| $(w_d, w_q)$ | **(0.65,0.35)** | **(0.6,0.4)** | **(0.57,0.43)** | **(0.55,0.45)** | **(0.53,0.47)** | **(0.5,0.5)** | **(0.45,0.55)** |
|---|---|---|---|---|---|---|---|
| Risk neutral | 0.7513 | 0.7709 | 0.7742 | 0.7717 | 0.7849 | 0.7592 | 0.7543 |

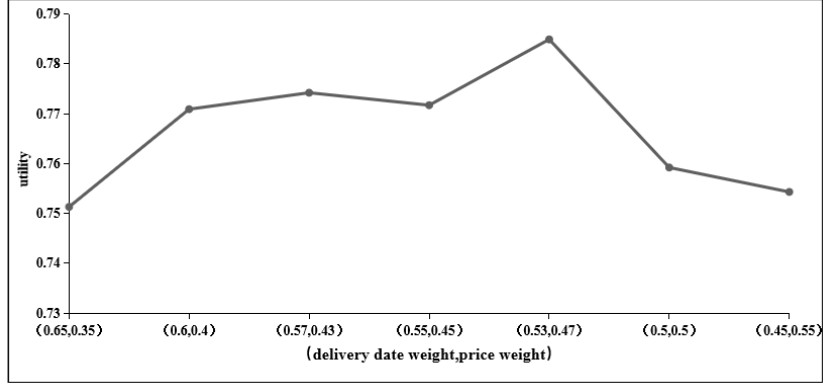

**Figure 6.** The utility of 4PL integrator under risk neutral.

Figure 5 shows that the highest utility of 4PL integrator under risk neutral is obtained at $w_d = 0.53$, $w_q = 0.47$. The highest utility of 4PL integrator for risk neutral auctions is obtained at $w_d = 0.53$, $w_q = 0.47$. Figure 6 shows that the highest utility of 4PL integrator under risk neutral is obtained at $w_d = 0.65$, $w_q = 0.35$. So, the highest utility of 4PL integrator under risk aversion and risk neutrality is not at complete information auctions, i.e., at $w_d = 0.55$, $w_q = 0.45$. So 4PL integrator will not get higher utility under complete information auctions. Therefore, the 4PL integrator can reserve the auction information appropriately before the auction.

## 5. Conclusions

This paper studies the 3PLselection problem in sustainable operations of 4PL management. 4PL integrator selects 3PL suppliers to complete the logistics project. A two-stage auction mechanism for 3PL supplier selection is designed. This issue is studied from the perspective of risk preferences and information disclosure levels of 4PL integrator. The results show that 4PL integrator retain the disclosure information properly before the auction in order to obtain higher utility; 4PL integrator will get higher utility in maintaining a risk-neutral rational attitude in auction.

There are still some limitations in this study. This paper only considers the risk aversion preference of 4PL integrator. In fact, 3PL suppliers also have risk aversion preferences. In future research, we will try to take into account the risk aversion of 3PL suppliers. The study will have an impact on the auction results when both 4PL integrator and 3PL suppliers are risk aversion. In terms of multiple attributes, this article only considers two attributes, i.e., delivery date and price. The attributes considered in practice will also include the quality and credibility of 3PL suppliers. Moreover, there is a correlation between the delivery date and the price in practice, and the correlation between the attributes can be taken into account in future studies. Finally, multi-attribute is considered in this paper, price and delivery date, which makes the selection process more complex. The combination of auctions with multi-criteria decision-making methods (MCDM) is an effective way to solve this problem. In this paper, MAUT is used, other methods like ELECTRE, MAVT, and TOPSIS can be considered in the future works.



**Author Contributions:** Conceptualization, F.L.; methodology, H.B.; software, W.F.; validation, F.L.; formal analysis, H.B.; investigation, Y.H.; data curation, Y.H.; writing—original draft preparation, W.F. and Y.H.; writing—review and editing, F.L.; visualization, H.B.; supervision, S.W. and X.Z.; project administration, F.L.; funding acquisition, F.L. and H.B. All authors have read and agreed to the published version of the manuscript.

**Funding:** This research was funded by the Key Technologies Research and Development Program of China under Grant No. 2020YFB1712802, the National Science Foundation of China under Grant No. 71401027, the Humanities and Social Sciences funds for Hebei Universities under Grant No. SQ202002, Natural Science Foundation of Hebei Province under Grant No. G2020030006.

**Institutional Review Board Statement:** Not applicable.

**Informed Consent Statement:** Not applicable.

**Conflicts of Interest:** The authors declare no conflict of interest.

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
