# Peer review of "A Two-Stage Auction Mechanism for 3PL Supplier Selection under Risk Aversion"

_sustainability, doi:10.3390/su13179745_

Round 1
Reviewer 1 Report
Dear authors, I don't think the line spacing is uniform. E.g. from line 26 to 48, it is smaller, and from line 48 onwards, it is larger. Please check. Line 97: capital letter ..«the paper exploits group decision-making and utility theory..« In my opinion, the introduction is too long and also contains content that would fall under the literature review. The introduction includes something general about the topic and answers the following questions: 1. Purpose, objectives and/or RQ. 2. Why choose this topic? Why should the topic interest the reader? 3. What is a scientific contribution? How is the article different from others with a similar theme. 4. Which existing article(s) has/have you upgraded? 5. Who is the content for? Everything except the fourth and fifth point is written in the introduction. PLEASE ADD. The current introduction section that does not fall under the above points should be written in the second chapter: A literature review. It is commendable that the authors made a comparison of the most commonly used methods for selecting logistics/suppliers. The authors should explain why they chose only two criteria (price and delivery time). They may indeed be the most important but by no means the only ones. The selection of a logistician based solely on these two criteria is misleading and covers only one pillar of sustainability. Please explain how the weights are determined? Which method is used to determine the weight? Which method can be used to determine the weight? In my opinion, the article is too long. The part describing the methodology should be more concise. The introductory part should be divided into two chapters: introduction and review of the literature. The disadvantages of this article are not the methodology, but: - the method of determining the weights, and - selection of a 3PL based on only two criteria. The authors selected only two criteria from the economic pillar of sustainability. There are many criteria for selecting a 3PL. It is also necessary to take into account qualitative criteria and at the same time not neglect any of the pillars of sustainability. I suggest that the authors expand the analysis and consider more criteria. In this case, we could talk about the added value of the article.Author Response
We would like to thank the editors and reviewers for their hard work and helpful suggestions, which are very important to improve the quality of the paper. We try our best to revise the paper according to the questions and suggestions, all of the revised parts are marked in blue. An illustration is given below.
Reply to the questions and suggestions (reviewer 1):
- I don't think the line spacing is uniform. E.g. from line 26 to 48, it is smaller, and from line 48 onwards, it is larger.
Reply: The space between lines are revised from line 48 onwards, now they have the same line spacing.
- Please check. Line 97: capital letter ..«the paper exploits group decision-making and utility theory.
Reply: Yes, t should be a capital letter, which has been revised.
- In my opinion, the introduction is too long and also contains content that would fall under the literature review. The introduction includes something general about the topic and answers the following questions: 1. Purpose, objectives and/or RQ. 2. Why choose this topic? Why should the topic interest the reader? 3. What is a scientific contribution? How is the article different from others with a similar theme. 4. Which existing article(s) has/have you upgraded? 5. Who is the content for? Everything except the fourth and fifth point is written in the introduction. PLEASE ADD. The current introduction section that does not fall under the above points should be written in the second chapter: A literature review.
Reply: We agree with the reviewer, and the Introduction has been divided into two parts: Introduction and Literature review. We have tried our best to restructure the content of the paper and answer the five questions.
- It is commendable that the authors made a comparison of the most commonly used methods for selecting logistics/suppliers. The authors should explain why they chose only two criteria (price and delivery time). They may indeed be the most important but by no means the only ones. The selection of a logistician based solely on these two criteria is misleading and covers only one pillar of sustainability.
Reply: Yes, there are many important criteria, that affect the selection of 3PL suppliers, such as quality, price, transaction cost, delivery time and so on. This paper mainly focuses on two important criteria, price and delivery time. One of the major purposes of the paper is to provide an efficient method to support the selection of 3PL suppliers. Therefore,
a two-stage auction mechanism is proposed. From the perspective of methodology, the method that can deal with two criteria, also has the ability to deal many criteria. At the beginning of writing the paper, we want to consider four criteria. However, the initial data is not enough. Finally, we consider two criteria, price and delivery time. In the future work, we will try to cover the gap.
- Please explain how the weights are determined? Which method is used to determine the weight? Which method can be used to determine the weight?
Reply: We enhanced the description about how to obtain the weights, revised the parts in section 4.2.3.
“Based on surveys of enterprises and literatures [37], the initial data about the delivery date weight and price weight are designed. Usually, the weights are value of points under the situation of complete information. However, the auction is under the situation of incomplete information, the values of the weights are not determined values or are uncertain values. Therefore, this section applies a form of interval to describe this kind of uncertainty, and then applies formula (2)-(4) to calculate the values of the delivery date weight and price weight, respectively.”
- In my opinion, the article is too long. The part describing the methodology should be more concise.
Reply: The paper is revised, especially, the methodology part. The number of pages is reduced from 26 to 19.
- The introductory part should be divided into two chapters: introduction and review of the literature.
Reply: We agree with the reviewer, and the Introduction has been divided into two parts: Introduction and Literature review.
- The disadvantages of this article are not the methodology, but: - the method of determining the weights, and - selection of a 3PL based on only two criteria. The authors selected only two criteria from the economic pillar of sustainability. There are many criteria for selecting a 3PL. It is also necessary to take into account qualitative criteria and at the same time not neglect any of the pillars of sustainability. I suggest that the authors expand the analysis and consider more criteria. In this case, we could talk about the added value of the article.
Reply: Thank you very much for your advices. Yes, there are many important criteria, that affect the selection of 3PL suppliers, such as quality, price, transaction cost, delivery time and so on. This paper mainly focuses on two important criteria, price and delivery time. One of the major purposes of the paper is to provide an efficient method to support the selection of 3PL suppliers. Therefore, a two-stage auction mechanism is proposed. From the perspective of methodology, the method that can deal with two criteria, also has the ability to deal many criteria. At the beginning of writing the paper, we want to consider four criteria. However, the initial data is not enough. Finally, we consider two criteria, price and delivery time. In the future work, we will try to cover the gap.
Reply to the questions and suggestions (reviewer 2):
- It should be justified why the paper corresponds to the topics of the Sustainability journal. The term “sustainability” is not mentioned in the paper. Even the Journal is not cited in the paper.
Reply: 3PL supplier selection in sustainable operation of fourth party logistics is an important part of sustainable supply chain management. Selecting good 3PL suppliers are very important to maintain sustainable operations of logistics businesses and supply chains. The two-stage auction mechanism provides a method support for fulfilling the goal of management. In the revised version of the paper, the aspect of sustainability is emphasized in abstract, introduction and other parts of the paper. Relative papers are cited in the context.
- Literature review is of insufficient quality and insufficient. It is necessary to cite more of the latest literature sources published in 2018-2021. Now, from the 42 sources only five of them are of this period. I suggest citing: (1) Jovic et al. 2019, Sustainability 11 (15) 4236. (2) Wen et al. 2019, Economic Research-Ekonomska Istraživanja 32 (1) 4033-4050. (3) Mohammad et al. 2021, Sustainability 13 (5) 2695. (4) Ebinger, Omondi 2020, Sustainability 12 (15) 6129. (5) Lu et al. 2021, TEDE 27 (2) 402-458.
Reply: We have tried many times to find out the five references listed above, but unfortunately, only the fourth reference has been found and cited. We need more and full information to find the other four. Please show us more information, thank you very much.
(4) Frank Ebinger, Bramwel Omondi, Leveraging Digital Approaches for Transparency in Sustainable Supply Chains: A Conceptual Paper, Sustainability, 2020, 12(15), 6129, 1-16

Reviewer 2 Report
I suggest accepting the paper with minor corrections.
The work contains a solid theoretical basis. The research design and instruments are valid and reliable. The paper can be fully understood by the targeted audience. The Abstract adequately summarizes the text. There is a clear description of a valid methodology. There is a clear demonstration of a valid process and analytical tools applied. The argumentation is clear to the reader. There is a case study developed in the paper. The paper is well structured. The paper is interesting and has theoretical and practical value.
However, minor corrections required:
It should be justified why the paper corresponds to the topics of the Sustainability journal. The term “sustainability” is not mentioned in the paper. Even the Journal is not cited in the paper.
Literature review is of insufficient quality and insufficient. It is necessary to cite more of the latest literature sources published in 2018-2021. Now, from the 42 sources only five of them are of this period.
I suggest citing:
Jovic et al. 2019, Sustainability 11 (15) 4236.
Wen et al. 2019, Economic Research-Ekonomska Istraživanja 32 (1) 4033-4050.
Mohammad et al. 2021, Sustainability 13 (5) 2695.
Ebinger, Omondi 2020, Sustainability 12 (15) 6129.
Lu et al. 2021, TEDE 27 (2) 402-458.
Author Response
We would like to thank the editors and reviewers for their hard work and helpful suggestions, which are very important to improve the quality of the paper. We try our best to revise the paper according to the questions and suggestions, all of the revised parts are marked in blue. An illustration is given below.
Reply to the questions and suggestions (reviewer 1):
- I don't think the line spacing is uniform. E.g. from line 26 to 48, it is smaller, and from line 48 onwards, it is larger.
Reply: The space between lines are revised from line 48 onwards, now they have the same line spacing.
- Please check. Line 97: capital letter ..«the paper exploits group decision-making and utility theory.
Reply: Yes, t should be a capital letter, which has been revised.
- In my opinion, the introduction is too long and also contains content that would fall under the literature review. The introduction includes something general about the topic and answers the following questions: 1. Purpose, objectives and/or RQ. 2. Why choose this topic? Why should the topic interest the reader? 3. What is a scientific contribution? How is the article different from others with a similar theme. 4. Which existing article(s) has/have you upgraded? 5. Who is the content for? Everything except the fourth and fifth point is written in the introduction. PLEASE ADD. The current introduction section that does not fall under the above points should be written in the second chapter: A literature review.
Reply: We agree with the reviewer, and the Introduction has been divided into two parts: Introduction and Literature review. We have tried our best to restructure the content of the paper and answer the five questions.
- It is commendable that the authors made a comparison of the most commonly used methods for selecting logistics/suppliers. The authors should explain why they chose only two criteria (price and delivery time). They may indeed be the most important but by no means the only ones. The selection of a logistician based solely on these two criteria is misleading and covers only one pillar of sustainability.
Reply: Yes, there are many important criteria, that affect the selection of 3PL suppliers, such as quality, price, transaction cost, delivery time and so on. This paper mainly focuses on two important criteria, price and delivery time. One of the major purposes of the paper is to provide an efficient method to support the selection of 3PL suppliers. Therefore,
a two-stage auction mechanism is proposed. From the perspective of methodology, the method that can deal with two criteria, also has the ability to deal many criteria. At the beginning of writing the paper, we want to consider four criteria. However, the initial data is not enough. Finally, we consider two criteria, price and delivery time. In the future work, we will try to cover the gap.
- Please explain how the weights are determined? Which method is used to determine the weight? Which method can be used to determine the weight?
Reply: We enhanced the description about how to obtain the weights, revised the parts in section 4.2.3.
“Based on surveys of enterprises and literatures [37], the initial data about the delivery date weight and price weight are designed. Usually, the weights are value of points under the situation of complete information. However, the auction is under the situation of incomplete information, the values of the weights are not determined values or are uncertain values. Therefore, this section applies a form of interval to describe this kind of uncertainty, and then applies formula (2)-(4) to calculate the values of the delivery date weight and price weight, respectively.”
- In my opinion, the article is too long. The part describing the methodology should be more concise.
Reply: The paper is revised, especially, the methodology part. The number of pages is reduced from 26 to 19.
- The introductory part should be divided into two chapters: introduction and review of the literature.
Reply: We agree with the reviewer, and the Introduction has been divided into two parts: Introduction and Literature review.
- The disadvantages of this article are not the methodology, but: - the method of determining the weights, and - selection of a 3PL based on only two criteria. The authors selected only two criteria from the economic pillar of sustainability. There are many criteria for selecting a 3PL. It is also necessary to take into account qualitative criteria and at the same time not neglect any of the pillars of sustainability. I suggest that the authors expand the analysis and consider more criteria. In this case, we could talk about the added value of the article.
Reply: Thank you very much for your advices. Yes, there are many important criteria, that affect the selection of 3PL suppliers, such as quality, price, transaction cost, delivery time and so on. This paper mainly focuses on two important criteria, price and delivery time. One of the major purposes of the paper is to provide an efficient method to support the selection of 3PL suppliers. Therefore, a two-stage auction mechanism is proposed. From the perspective of methodology, the method that can deal with two criteria, also has the ability to deal many criteria. At the beginning of writing the paper, we want to consider four criteria. However, the initial data is not enough. Finally, we consider two criteria, price and delivery time. In the future work, we will try to cover the gap.
Reply to the questions and suggestions (reviewer 2):
- It should be justified why the paper corresponds to the topics of the Sustainability journal. The term “sustainability” is not mentioned in the paper. Even the Journal is not cited in the paper.
Reply: 3PL supplier selection in sustainable operation of fourth party logistics is an important part of sustainable supply chain management. Selecting good 3PL suppliers are very important to maintain sustainable operations of logistics businesses and supply chains. The two-stage auction mechanism provides a method support for fulfilling the goal of management. In the revised version of the paper, the aspect of sustainability is emphasized in abstract, introduction and other parts of the paper. Relative papers are cited in the context.
- Literature review is of insufficient quality and insufficient. It is necessary to cite more of the latest literature sources published in 2018-2021. Now, from the 42 sources only five of them are of this period. I suggest citing: (1) Jovic et al. 2019, Sustainability 11 (15) 4236. (2) Wen et al. 2019, Economic Research-Ekonomska Istraživanja 32 (1) 4033-4050. (3) Mohammad et al. 2021, Sustainability 13 (5) 2695. (4) Ebinger, Omondi 2020, Sustainability 12 (15) 6129. (5) Lu et al. 2021, TEDE 27 (2) 402-458.
Reply: We have tried many times to find out the five references listed above, but unfortunately, only the fourth reference has been found and cited. We need more and full information to find the other four. Please show us more information, thank you very much.
(4) Frank Ebinger, Bramwel Omondi, Leveraging Digital Approaches for Transparency in Sustainable Supply Chains: A Conceptual Paper, Sustainability, 2020, 12(15), 6129, 1-16
Reviewer 3 Report
The authors presented an interesting study in the field of 3PL supplier selection in terms of 4PL. It is proposed to use auctions as an additional selection tool, as well as to consider the attitude to risk.
In our opinion, the combination of such approaches is quite productive.
The research contains scientific novelty. The manuscript is well-structured and is likely to be of interest to readers.
1. The authors correctly identified in the conclusion some limitations of the proposed method, as well as ways to overcome these limitations in future studies. However, I propose to pay attention to the more common approach to sourcing for sustainable supply chains, based on the use of multi-criteria decision-making methods (MCDM). The authors mentioned several studies in this area. However, in my opinion, the combination of auctions with MCDM is one of the promising directions for the development of this study. I suggest supplementing the conclusion with the authors' thoughts on this important issue. In addition, it is advisable to mention the related research doi: 10.3390/sym9090169 in the literature review.
2. The legislation of various countries is a serious limitation to the implementation of the proposed approach. Suppliers as part of supply chains or international transport corridors may be regulated by the legislation of the respective countries. I suggest discussing the procedure for assessing risks in the proposed method, considering this and similar constraints (political, climatic).
3. Finally, I suggest expanding the literature review with important research in the field of auctions: Mithas S., Jones J. Do Auction Parameters Affect Buyer Surplus in E-Auctions for Procurement? Production and Operations Management. 2007, 4, 455-470.
Author Response
We would like to thank the editors and reviewers for their hard work and helpful suggestions, which are very important to improve the quality of the paper. We try our best to revise the paper according to the questions and suggestions, all of the revised parts are marked in blue. An illustration is given below.
- The authors correctly identified in the conclusion some limitations of the proposed method, as well as ways to overcome these limitations in future studies. However, I propose to pay attention to the more common approach to sourcing for sustainable supply chains, based on the use of multi-criteria decision-making methods (MCDM). The authors mentioned several studies in this area. However, in my opinion, the combination of auctions with MCDM is one of the promising directions for the development of this study. I suggest supplementing the conclusion with the authors' thoughts on this important issue. In addition, it is advisable to mention the related research doi: 10.3390/sym9090169 in the literature review.
Reply: The conclusion is revised, some advice about combination of auction and MCDM are introduced, see section “Conclusion”. The reference “Edmundas Kazimieras Zavadskas, Dragan Pamucar, Zeljko Stevic, Abbas Mardani, Multi-Criteria Decision-Making Techniques for Improvement Sustainability Engineering Processes, MDPI, 2020” is cited and referenced in the paper.
- The legislation of various countries is a serious limitation to the implementation of the proposed approach. Suppliers as part of supply chains or international transport corridors may be regulated by the legislation of the respective countries. I suggest discussing the procedure for assessing risks in the proposed method, considering this and similar constraints (political, climatic).
Reply: Thank you very much for your advises. Legislation, political and climatic are important factors, which will impact the suppliers in the supply chain. But this factors have different attributes with the factors in this paper, like price or delivery date, they are all factors in the level of operation. Therefore, we would like to invite some experts on Legislation, political or climatic to make a cooperative research in the next step of our work. Thanks again for your kind advises.
- Finally, I suggest expanding the literature review with important research in the field of auctions: Mithas S., Jones J. Do Auction Parameters Affect Buyer Surplus in E-Auctions for Procurement?,Production and Operations Management. 2007, 4, 455-470.
Reply: The paper has been cited and analyzed in section “2.1 Auction mechanism and information disclosure”.
